# Genomic Landscape of Susceptibility to Severe COVID-19 in the Slovenian Population [note 1]

**DOI:** 10.3390/ijms25147674

**Published:** 2024-07-12

**Authors:** Anja Kovanda, Tadeja Lukežič, Aleš Maver, Hana Vokač Križaj, Mojca Čižek Sajko, Julij Šelb, Matija Rijavec, Urška Bidovec-Stojković, Barbara Bitežnik, Boštjan Rituper, Peter Korošec, Borut Peterlin

**Affiliations:** 1Clinical Institute of Genomic Medicine, University Medical Centre Ljubljana, 1000 Ljubljana, Slovenia; 2Faculty of Medicine, University of Ljubljana, 1000 Ljubljana, Slovenia; 3University Clinic of Respiratory and Allergic Diseases Golnik, 4204 Golnik, Slovenia

**Keywords:** severe COVID-19, severe outcome of SARS-CoV-2 infection, whole-genome sequencing, WGS, genetic susceptibility, rare variants, human rare genomic variants

## Abstract

Determining the genetic contribution of susceptibility to severe SARS-CoV-2 infection outcomes is important for public health measures and individualized treatment. Through intense research on this topic, several hundred genes have been implicated as possibly contributing to the severe infection phenotype(s); however, the findings are complex and appear to be population-dependent. We aimed to determine the contribution of human rare genetic variants associated with a severe outcome of SARS-CoV-2 infections and their burden in the Slovenian population. A panel of 517 genes associated with severe SARS-CoV-2 infection were obtained by combining an extensive review of the literature, target genes identified by the COVID-19 Host Genetic Initiative, and the curated Research COVID-19 associated genes from PanelApp, England Genomics. Whole genome sequencing was performed using PCR-free WGS on DNA from 60 patients hospitalized due to severe COVID-19 disease, and the identified rare genomic variants were analyzed and classified according to the ACMG criteria. Background prevalence in the general Slovenian population was determined by comparison with sequencing data from 8025 individuals included in the Slovenian genomic database (SGDB). Results show that several rare pathogenic/likely pathogenic genomic variants in genes *CFTR*, *MASP2*, *MEFV*, *TNFRSF13B*, and *RNASEL* likely contribute to the severe infection outcomes in our patient cohort. These results represent an insight into the Slovenian genomic diversity associated with a severe COVID-19 outcome.

## 1. Introduction

The ongoing coronavirus pandemic, caused by severe acute respiratory syndrome coronavirus 2 (SARS-CoV-2), results in coronavirus disease 19 (COVID-19) [1,2,3], that manifests with a wide range of symptoms ranging from asymptomatic to severe. Patients present with symptoms such as fever, cough, fatigue, headache, hemoptysis, diarrhea, dyspnea, lymphocytopenia, pneumonia, acute respiratory distress syndrome, and acute cardiac injury, and present with specific radiological findings of ground-glass opacities. Severe cases require hospitalization and may lead to death [4].

Being an RNA virus, SARS-CoV-2 continuously evolves both by mutation and recombination during the replication of the genome, and since its first detection, several lineages, sub-lineages, and variants have evolved. Despite some variants evading immunity from previous infection or vaccination, the variability in the clinical manifestation (from asymptomatic to severe) of infection cannot be explained by the viral variability alone, and several human host factors, such as age, sex, and various comorbidities are now known to play a role [5,6]. Similarly, the immunological effects and their role in disease pathogenesis have now also been well characterized [7].

However, the underlying cause remains to be determined, and severe manifestations of infection, including death in otherwise healthy middle-aged individuals, suggest that genetic factors may play a role [8]. Indeed, because of genetics’ potential importance in disease prognosis, prevention, and public health planning measures, the field of COVID-19-host genetics has expanded rapidly, resulting in several thousand publications on this topic in the past four years. Disease-associated population-specific variant information is particularly important as it is prerequisite for any future development of screening tests to identify high-risk individuals.

The results of this research have been more complex. Following early reports on the association between blood groups and the occurrence of infection [9,10], and an initial examination of a handful of candidate genes, based on their association with other viral interactions, such as *ACE2*, *CLEC4M*, *MBL*, *ACE*, *CD209*, *FCER2*, *OAS-1*, *TLR4*, and *TNF-*α [11,12,13], in the years since, several hundred genes have been implicated to play a role in the complex host genetic contribution to COVID-19. Genome-wide association studies (GWAS) have shown that many common genomic variants are enriched in cohorts of patients with severe COVID-19 [14,15]. So far, the identified polymorphisms show geographical differences, and were mostly shown to have weak effects, and even combining them to assess their polygenic risk score (PRS) has not so far been able to effectively predict disease outcome. Similarly, while it has been proposed that the accumulation of weak effects of many rare functional variants may contribute to the overall risk in patients with severe disease [16,17], the information on such variants is scarce in many populations, including ours. 

Therefore, we aimed to determine rare genomic variants and their burden in a comprehensive set of evidence-based genes in well-characterized patients hospitalized due to COVID-19 in Slovenia.

## 2. Results

We sequenced the whole genomes of 60 patients, hospitalized during the second pandemic wave of COVID-19 in Slovenia [7], to identify variants of interest in 517 evidence-based genes associated with severe COVID-19. This evidence-based gene panel was constructed by combining three strategies: literature curation of original research publications; high-impact targets identified by the GWAS COVID-19 Host Genetic Initiative (COVID-19 HGI) [18]; and high-support genes from a research COVID-19 associated gene panel (COVID-19 research v1.136 PanelApp), curated by EnglandGenomics [19] (Figure 1). The identified variants were classified according to American College of Medical Genetics and Genomics (ACMG) criteria [20] and the model of inheritance. The resulting pathogenic/likely pathogenic variants and risk factors, as well as their Slovenian genomic database (SGDB) background prevalence, are given in Table 1.

In total, we identified variants of interest (pathogenic/likely pathogenic variants or risk factors) in 52 of the 60 hospitalized patients. Of the 52 patients with variants of interest, 25 (48%) presented with a single variant, and 27 (52%) presented with more than one variant (Figure 2), highlighting the complex genetics of immunological response to viral infections and the difficulty in assigning causality. 

Despite including hundreds of genes in our evidence-based panel, by adhering to strict ACMG criteria, we finally classified a total of 32 variants as pathogenic/likely pathogenic and 4 as risk factors, in 27 of the genes included in the analysis (Table 1). No variants of interest were identified in the *LZTFL1* gene, other than the two previously reported GWAS-associated variants rs17713054 and rs35482426 (Table 1). The estimated burden of all identified pathogenic/likely pathogenic variants of interest in our COVID-19 hospitalized patients compared to the control Slovenian population was statistically significant (*p* = 2.8 × 10^−5^).

Of these 36 variants of interest, 9 were previously reported as pathogenic/likely pathogenic in the ClinVar database [21], while 17 were reported with conflicting interpretations (unrelated to COVID-19) that included at least one pathogenic/likely pathogenic report. A further six of the variants were unclassified in the ClinVar database and were classified as likely pathogenic for the first time in this study. Of the 35 patients with pathogenic/likely pathogenic variants, variants in 10 patients were consistent with the proposed inheritance model, while 25 patients were heterozygous for variants originally associated with an autosomal-recessive disease. 

The 7 out of 32 identified pathogenic/likely pathogenic variants that fit the inheritance model proposed for their respective five genes, were identified in *CFTR* (two variants in possible compound heterozygosity), *MASP2* (one homozygous occurrence), *MEFV*, *RNASEL*, and *TNFRSF13B.* Pathogenic/likely pathogenic variants consistent with the proposed inheritance model were identified in 10 patients in total (Table 1 and Table 2).

Interestingly, out of the 7 patients who died, 3 patients had pathogenic/likely pathogenic variants, consistent with predicted inheritance, vs. 7 in 53 patients who survived severe COVID-19; however, this was not statistically significant. Two of the deceased patients had likely pathogenic variants in the *MEFV* gene, and one patient had a homozygous pathogenic variant in *MASP2*. Of the remaining four patients who died, one had a risk factor in the *MBL2* gene and one patient in the *PRF1* gene, one had a pathogenic variant in the AR gene *IL36RN*, and no variants of interest were identified in one of the deceased patients. 

Of the remaining seven patients with pathogenic/likely pathogenic variants who were hospitalized but survived, four had likely pathogenic variants in *MEFV*, one was compound heterozygous for pathogenic/likely pathogenic variants in *CFTR*, and one each had a pathogenic variant in *RNASEL* and *TNFRSF13B*, respectively. Interestingly, most had additional risk factors (Table 2). Indeed, most patients had several variants of interest (Figure 2), but interestingly, the absence of such variants of interest did not show a statistically significant association with survival (χ^2^ test *p* = 0.937137). 

In total, 35 of the 60 patients carried more than 48 risk factor variants in *MBL2* (34), *PRF1* (7), and *APOE* (7) (Table 2). The *LZTFL1* risk factors rs17713054 and rs35482426 were present in a total of 16 patients, and were the only finding in 2 patients (Table 2). 

In addition to strong variants of interest, we also identified more than 300 variants of uncertain significance in more than 200 genes from our curated evidence-based panel (Appendix A). These variants include those already classified as variants of uncertain significance in the ClinVar database or classified as such in this study due to being ultra-rare and having at least a moderate coding impact but lacking other criteria for pathogenicity. Most of these variants were missense variants that would need a functional assessment to reach a pathogenicity classification. Their significance remains to be determined in further studies.

## 3. Discussion 

The significant effort invested in studying COVID-19-host genetics by research groups worldwide has resulted in numerous insights into the pathogenesis of the disease from different fields, from genetics to immunology, highlighting the complexity of virus–host interactions. It is now known that COVID-19-host genetics is very complex and that disease progression and outcome may correlate with more than one gene or genomic region. Indeed, it is estimated that many hundreds of genes are involved and have a direct or indirect effect on the clinical course of COVID-19 infection. In our study design, we have therefore made an effort to include in our analysis all genes for which the highest level of evidence was gathered. Applying stringent interpretation criteria to the identified variants yielded a total of seven pathogenic/likely pathogenic variants consistent with the proposed inheritance model in 10 patients in total, in five genes: *CFTR* (two variants in possible compound heterozygosity), *MASP2* (one homozygous occurrence), *MEFV*, *RNASEL*, and *TNFRSF13B.* The allelic frequency of the variants in these genes in the Slovenian population, based on >8000 individuals, is given in the Table 1 column ‘SGDB allelic frequency’. While *MASP2* and *MEFV* are relatively common, variants in *CFTR*, *RNASEL* and *TNFRSF13B* are rare, which is likely linked to their expressivity and disease penetrance.

The *CFTR* gene product, cystic fibrosis transmembrane conductance regulator, is an ATP-binding cassette transporter that functions as a ligand-gated anion channel involved in epithelial ion transport. Biallelic pathogenic variants in the *CFTR* gene cause cystic fibrosis [22], manifesting in chronic bronchopulmonary dysfunction. The penetrance of CFTR is complex, ranging from autosomal dominant in the case of bronchiectasis (OMIM: 211400) to autosomal recessive in the case of cystic fibrosis, with expressivity being variant-dependent. For example, in our cohort, one patient was found to be a possible compound heterozygote for likely pathogenic variants in the *CFTR* gene, while five patients were found to be carriers of pathogenic/likely pathogenic variants in the heterozygous state. Of note, the intronic *CFTR*:c.1210-11T>G variant identified in our study, while considered pathogenic, is known to only be so in the compound heterozygous state with severe pathogenic *CFTR* variants, while homozygous individuals are asymptomatic [23]. However, apart from this variant, carriers of single cystic fibrosis-causing variants of the *CFTR* gene were previously also shown to be more susceptible to the severe form of COVID-19 [24].

The *MASP2* gene encodes for a mannan-binding lectin serine protease. Biallelic pathogenic variants in this gene are an established cause for autosomal recessive MASP2 deficiency, manifesting with increased susceptibility to infection due to the defective activation of the complement system [25,26]. *MASP2* expressivity, measured by its plasma level, varies in different populations [27] and its penetrance is highly variable, with an estimated 5–18% of MASP2-deficient individuals remaining asymptomatic [27,28]. In our cohort, we identified one likely pathogenic variant in *MASP2* in eight patients, in seven patients in heterozygous form and in one patient in homozygous form. 

Pathogenic and likely pathogenic heterozygous variants in the *MEFV* gene, which encodes an innate immune sensor, lead to the production of inflammatory mediators during infection [29], and are an established cause of autosomal dominant familial Mediterranean fever. The main expression of MEFV occurs in mature granulocytes, with an alternative isoform expressed in leukocytes. The penetrance of *MEFV* pathogenic variants is complex, ranging from asymptomatic heterozygotes to heterozygotes with a clinical manifestation of Mediterranean fever. In the Slovenian population, the identified K695R variant is relatively common in apparently healthy individuals, as evident from its SGDB prevalence and previous studies examining the carrier rate in Slovenia [30]. Interestingly, it was shown that infected Mediterranean fever patients do not have a worse outcome of SARS-CoV-2 infection when already hospitalized [31].

RNASEL is involved in intracellular single RNA cleavage as part of the dsRNA detection system and antiviral response [32,33], but the dysfunction of RNASEL also increases the risk of cancer and autoimmune disease [34] in an apparently autosomal dominant manner. A pathogenic variant in the *RNASEL* gene, encoding a 2-5A-dependent RNase, was found in one patient. Apart from SARS-CoV-2, RNASEL has so far been identified to play an antiviral role against the Dengue virus [35]. 

Finally, a pathogenic heterozygous variant in the *TNFRSF13B* gene, whose product plays a crucial role in humoral immunity [36], was identified in one of our patients. The heterozygous, homozygous, and compound heterozygous variants in *TNFRSF13B* lead to common variable immunodeficiency (CVID) or selective IgA deficiency (OMIM: 240500), that are characterized by susceptibility to recurrent infections with many different infectious agents [37]. Pathogenic variants in the *TNFRSF13B* were also previously identified in isolated cases of patients with severe COVID-19 [38,39].

While the gene-associated phenotype and the proposed inheritance model were both considered as parts of the ACMG criteria, it is important to note that the association of a particular variant with SARS-CoV-2 infection outcome may not follow the same inheritance model as the original association of the gene with another disease in which this gene is involved. Therefore, although the 26 heterozygous variants in the *ADAR*, *AIRE*, *ATM*, *BRCA2*, *C2*, *C6*, *C7*, *C8B*, *C9*, *CFD*, *CFI*, *CFTR*, *CYBA*, *FANCC*, *HAVCR2*, *HPS4*, *IL36RN*, *MASP2*, *PGM3*, *POLR3A* and *RNU4ATAC* genes cannot be automatically considered to have an effect due to their otherwise autosomal-recessive association with their respective diseases, their contribution to SARS-CoV-2 infection severity would need to be functionally assessed or validated by additional studies. Until such studies provide a final confirmation, their contribution to the severe manifestation of COVID-19 remains unknown (Table 1 and Table 2). For example, it is known that pathogenic heterozygous variants in the *CFTR* gene lead to an increased risk of chronic pancreatitis, atypical mycobacterial infections, and bronchiectasis [40], and in the future some heterozygous variants in AR genes may be shown to confer an increased risk in case of COVID-19 severity.

Also in line with previous research, which has shown that many different genetic factors can concurrently contribute to the course of COVID-19 infection [16,17], in our study, we also identified more than one pathogenic/likely pathogenic variant or risk factor in almost half of the patient cohort. In particular, pathogenic/likely pathogenic variants that would not be sufficient for clinical manifestation according to the proposed inheritance model were found in 42% of patients, which raises an interesting question on their biological relevance. Indeed, similarly to other groups, we observed a clear burden of rare genomic variants classified as pathogenic/likely pathogenic in genes involved in immunity and host defense, autoinflammation, and autoimmunity. These variants were enriched in the selected patient cohort with severe infection outcomes compared to the Slovenian population [41]. 

Additionally, we identified risk factor variants in 58% (35/60) of our patients, while they were the sole finding in 28% (17/60) of our patient cohort. A total of four risk factor variants were identified in the following three genes; *APOE*, *MBL2*, and *PRF1*. 

The *APOE4* risk factor variant (NM_000041.4(*APOE*):388T>C, NM_001302688.2(*APOE*):c.466T>C, synonyms APOE4: C130R, C156R, and C112R) (rs429358) was identified in seven patients (12%), with one patient found to be homozygous for this allele. This risk factor is generally associated with hyperlipoproteinemia, cardiovascular disease, and Alzheimer’s disease and has shown an association with COVID-19 disease in several independent studies [42,43,44]; however, as hyperlipoproteinemia, cardiovascular disease, and dementia independently predict a worse outcome, the specific mechanism of its SARS-CoV-2 association remains elusive.

Two common *MBL2* risk factors, increasing COVID-19 susceptibility due to MBL deficiency [45,46,47], were identified in 32 patients in total (53%). Although MBL deficiency is considered an autosomal dominant disease, similar to MASP2 deficiency, it is a common deficiency, and patients may remain asymptomatic. In two of these patients, both variants, *MBL2*:c.154C>T (rs5030737) and *MBL2*:c.161G>A (rs1800450) located in trans, were detected, while two patients were found to be homozygotes for the *MBL2*:c.161G>A variant. 

Finally, the risk factor *PRF1*:c.272C>T variant (rs35947132) was detected in seven patients (12%). *PRF1* gene encodes Perforin-1, a pore-forming protein homologous to complement component C9 with a similar mechanism of transmembrane channel formation [48]. While pathogenic variants in the *PRF1* gene are associated with aplastic anemia, autosomal recessive familial hemophagocytic lymphohistiocytosis, and non-Hodgkin lymphoma, the c.272C>T risk factor variant (rs35947132) was previously also found to be enriched in several different COVID-19 patient cohorts [49,50,51]. 

Interestingly, we did not identify any variants of interest in the *LZTFL1* gene that is located within the 3p21.31 risk “Neanderthal” haplotype, identified as important by large GWAS studies [14,52], and carried by approximately 16% of Europeans [53]. Therefore, we also examined the frequency of the two previously reported intergenic variants surrounding the *LZTFL1* gene, rs17713054 [14,52] and rs35482426 [14], in the patient cohort and compared it with that of the Slovenian population. There was no difference in the observed allelic frequency of the rs17713054 intergenic variant and the rs35482426 intronic variant between our patient cohort and the general Slovenian population (Table 1).

While our patient cohort was heterogeneous with high comorbidities, this is a typical scenario observed in clinical practice when hospitalization is required for COVID-19. In such patients, by using an evidence-based gene panel and ACMG criteria for classification, pathogenic/likely pathogenic genomic variants were identified in 17% (10/60) of the patients. The presence of pathogenic/likely pathogenic variants consistent with the inheritance model was statistically significantly related to a further negative outcome. However, patients with no identified variants were not less likely to die, limiting our ability to draw any conclusion based on this small number of patients. 

### Limitations

Our work has the following inherent limitations, which are shared with other similar studies involving control and patient group composition, sample size, and limitations of variant classification. The control group consisted of individuals with unknown COVID-19 infection outcomes. Despite initially aiming to include a control group with mild or absent infection outcome, during the study, the occurrence of multiple infections as well as different vaccination statuses prevented us from obtaining such a group (i.e., vaccinated individuals may have a mild phenotype despite carrying a genetic variant conferring risk of a severe COVID-19 outcome). Similarly, to avoid any inherent bias, the patients hospitalized due to COVID-19 were selected in a blinded fashion regarding their immunological status (as determined in the original immunological study to which they were recruited). Despite these measures, the relatively small sample size of the severe COVID-19 cohort limits our ability to detect all rare variants in our population. Furthermore, the ACMG classification criteria limited our ability to classify variants as pathogenic to mostly exonic variants, where their pathogenicity could be assessed based on their predicted outcome on the protein level without further functional studies. Finally, since we did not perform phasing of variants, the compound heterozygosity of the variants in genes with the proposed autosomal recessive inheritance model is assumed but needs additional confirmation. Altogether, genomic analysis identified pathogenic/likely pathogenic variants in the *CFTR*, *MASP2*, *MEFV*, *RNASEL*, and *TNFRSF13B* genes in 10 patients included in this study. As these genes are associated with immunodeficiency, susceptibility to infections, and inflammatory disorders, and were consistent with the proposed inheritance model for the gene, these variants could likely be major contributors to the severity of SARS-CoV-2 infection in the case of these 10 patients. On the other hand, more than one-third of patients carried a single pathogenic/likely pathogenic variant in at least one COVID-19-associated gene with the proposed autosomal recessive inheritance model. For these variants, there is a possibility that they are additionally contributing to the severe condition of patients with COVID-19, which is a multifactorial disease. While it would be interesting to investigate the pathogenicity of intronic and other non-coding variants in genes with autosomal recessive inheritance model in patients where already one pathogenic/likely pathogenic variant or variant of uncertain significance was identified, this is unfortunately beyond the scope of our current research.

## 4. Materials and Methods

### 4.1. Patients

The severe COVID-19 cohort consisted of 60 patients hospitalized during September–December 2020 (the second pandemic wave in Slovenia) at the University Clinic of Respiratory and Allergic Diseases, Golnik, Slovenia, that participated in the Immunological Characteristics of COVID-19 Patients clinical trial (registered at ClinicalTrials.gov (NCT04679428) [7]). Patients or their legally authorized representatives provided informed consent to participate in the study (Slovenian National Medical Ethics Committee Approval No. 0120-201/2020/7 and 0120-333/2020/3).

Hospitalization admission criteria required supplemental oxygen at admission and/or radiological signs of COVID-19 pneumonia. Additional hospitalization criteria included age (above 65), body mass index (above 30), the presence of chronic kidney, cardiovascular or lung disease, diabetes, cancer, advanced liver disease, and the presence of immune insufficiency, as described in detail in the original trial publication [7]. Table 3 shows the demographic and clinical data of the included hospitalized patients. To exclude unintentional bias, the patients included in the current study belonged to all six immune-phenotype groups identified and described in the original study (i.e., they were chosen in a blinded fashion from the original immunological study).

### 4.2. Controls

Controls consisted of whole genome- or whole exome (where appropriate)-derived variants obtained from 8025 de-identified healthy individuals included in the SGDB at the Clinical Institute of Genomic Medicine (CIGM), University Medical Centre Ljubljana (UMCL), Slovenia (last accessed on 15 May 2024). All included individuals gave informed consent for the de-identified use of their variant information for research purposes. The informed consent statement was prepared according to National Review Board guidelines and approved by the Institutional Ethics Board at the UMCL, Slovenia. Control individuals were used only to determine background variant prevalence. The COVID-19 infection outcome status of the healthy individuals was unknown.

### 4.3. Whole Genome Sequencing

Whole genome sequencing was performed as previously described [54]. Briefly, third-party sequencing center services were used, where a standardized sequence of procedures was performed, consisting of PCR-free WGS library preparation protocol Illumina TrueSeq DNA Nano and sequencing on Illumina NovaSeq 6000 platform (both manufactured by Illumina, San Diego, CA, USA). The mean autosomal depth was greater than 30×. Data analysis, including variant calling, was performed in house at the CIGM, using the Genome Analysis Toolkit Best Practices workflow from the Broad Institute [54,55,56]. The dbscSNV database of precomputed splice effect predictions and the SpliceAI algorithm were used to detect and annotate splice site variants [57,58]. 

### 4.4. Gene Panel Selection Strategy

Three strategies were used to select an evidence-based gene panel associated with severe SARS-CoV-2 infection: literature curation of original research publications; high-impact targets identified by the GWAS COVID-19 Host Genetic Initiative (COVID-19 HGI) [18]; and high-support genes from a research COVID-19 associated gene panel (COVID-19 research v1.136 PanelApp), curated by EnglandGenomics [19] (Figure 1). 

### 4.5. Literature Curation

To identify human genes and genomic variants associated with SARS-CoV-2 infection, PubMed (www.ncbi.nlm.nih.gov/pubmed), and PubMedCentral (https://www.ncbi.nlm.nih.gov/pmc/) searches were performed from March 2021 to May 2022 using the following search strings: (<((genetic variation*) OR (gene polymorphism*) OR (polymorphism*) OR (gene variant*) OR (allele*) OR(GWAS*) OR (variant*) OR (host genetics*) OR (genetic susceptibility*) OR (genome*wide association study*)) AND ((COVID-19*) OR (SARS-CoV-2*) OR (coronavirus*)) > [[Title/Abstract]]).

We aimed to include only original research reports, such as observational, cross-sectional, and/or cohort studies exploring the link between the disease progression and or severity and their connection to the genetic variations found in the affected population. The individuals included in these studies had to have a clinical diagnosis of COVID-19 infections (positive PCR or antigen test) and their symptoms had to be professionally assessed. Patient outcomes had to be defined in the studies (mortality, severity, disease symptoms, etc.). The genomic markers had to be defined according to the European Medicine Agency definition [59] and included single-nucleotide polymorphisms (SNPs), short sequence repeat variants, haplotypes, alternative alleles, and other variants. 

We only included articles in the English language. The following exclusion criteria were used to screen the publications’ title/abstracts: duplicates, clinical drug trials, in vitro assays, in silico studies, animal studies, population studies, reviews, and meta-analyses were excluded, as were studies reporting results from fewer than 30 human subjects/cases. In cases where reviews referenced original work missed by our search string strategy, these publications were also included in the analysis (Figure 1).

Next, the publications were analyzed if they qualified for inclusion regarding participants, interventions, controls, and outcomes (PICO) [60], as detailed in Appendix A. The selected articles had to meet at least three of the four PICO criteria. Due to the volume of the literature regarding COVID-19 since 9 May 2022, to cross-check that no important studies were missed, top-journal review references after 9 May 2022 were examined for additional original functional studies. Only genes/variants replicated by at least one independent study, or results confirmed functionally were included in the final selection of genes/genomic variants. In this way, 170 original studies of human genetic variation related to SARS-CoV-2 were included in the screening (Figure 1, Appendix A).

### 4.6. COVID-19 Human Genetics Initiative 

The COVID-19 HGI was established in 2020 to facilitate the collaboration of the human genetics scientific community in COVID-19 host-genetics research [18]. The initiative has so far generated important results supporting the role of several human genetic variants in COVID-19 [14,61,62,63,64,65,66,67]. We have therefore also included the results generated by the COVID-19 HGI in our final gene panel selection (Figure 1, Appendix A).

### 4.7. Genomics England PanelApp

The Genomics England PanelApp is a crowdsourcing tool for gene panels, where they can be shared, downloaded, and evaluated by the scientific community [19]. Their COVID-19 research panel (current version 1.136) currently contains a set of 697 entities, of which 461 (459 genes and 2 regions) have been curated by expert reviewers, as having good evidence for association with COVID-19. The criteria for this curation include, but are not limited to, the following: there being multiple original studies demonstrating such association; a clinical study of an association that is supported by additional validation (in vitro/in vivo/in silico/other evidence); research and/or clinical community consensus on the likely importance of this gene; previous strong evidence of the involvement of the gene in a primary immunodeficiency disorder; and the gene being on the current list of known human genes associated with inborn errors of immunity by the International Union of Immunological Societies (IUIS) [19,68]. These 459 genes were also included in our final gene panel selection (Figure 1, Appendix A).

### 4.8. Genes and Interpretation

The final selection of genes in our SARS-CoV-2 gene panel included 517 genes. Of these genes, the following 79 were identified from at least two independent sources, were functionally confirmed, and/or were identified through the joint efforts of the COVID-19 HGI: ***ABO***, ***ACE2***, *ACSL6*, *AGT*, *AGTR1*, *AK5*, *APOE*, *ARHGEF38*, ***ATP11A***, *BCL11A*, *CASC20*, ***CCHCR1***, ***CCR2***, *CCR5*, ***CCR9***, *CFTR*, *CXCR6*, ***DPP9***, *EFNA4*, *ELF5*, ***FBRSL1***, ***FOXP4***, *FURIN*, ***FUT2***, *FYCO1*, *HCN3*, *HIP1*, *HLA-DRB1*, *IFITM3*, *IFNA10*, *IFNAR1*, ***IFNAR2***, *IFNL3*, *IFNL4*, *IL10RB*, *IL6*, *IRF1*, *IRF3*, *IRF7*, *JAK1*, ***KANSL1***, *LTA*, ***LZTFL1***, *MBL2*, ***MUC5B***, *NOS3*, *NR1H2*, *NXPE3*, ***OAS1***, *OAS2*, ***OAS3***, *PLEKHA4*, ***PLSCR1***, *PNPLA3*, *PRF1*, *RAB2A*, ***RAVER1***, *RGMA*, *SFTPD*, *SLC22A31*, *SLC2A5*, ***SLC6A20***, *SRRM1*, *TBK1*, *THBS3*, *TICAM1*, *TLL1*, *TLR3*, *TLR4*, *TLR7*, *TMEM65*, ***TMPRSS2***, *TNF*, *TRIM46*, ***TYK2***, *VDR*, ***XCR1***, *ZGLP1*, and *ZKSCAN1*. The genes in bold are those identified by international GWAS and multiple independent publications. The full list of the 517 genes is given in Appendix A.

The background population frequency of the identified variants of interest was determined from the GnomAD [69] and the SGDB (CIGM, UMCL). Genomic data interpretation was performed according to the ACMG/AMP standards and guidelines [20] and was performed by at least two analysts of whom at least one was a specialist in clinical and/or laboratory genetics, as previously described [56]. 

The proposed inheritance model and disease mechanisms of each gene were taken into account in the interpretation. Finally, the previously reported clinical manifestations associated with each identified variant were assessed in regard to their relevance to COVID-19-associated disease. All classified variants of interest are listed in Appendix A. 

### 4.9. Statistical Data Analysis

Statistical data analysis was performed using version 4.1.3 of the R computing environment [70]. A *p*-value of <0.05 was considered statistically significant. Genotype and allele frequencies were compared between the case and control group using the chi-square test or Fisher’s exact test when the expected frequency was less than 5. In controlling the false discovery rate (FDR) for multiple testing, the Benjamini and Hochberg adjustment of *p*-values was employed [71]. A burden analysis was conducted to determine if there was a higher prevalence of all identified pathogenic/likely pathogenic variants of interest in our COVID-19 hospitalized patients compared to the general Slovenian population. We estimated the burden using the chi-square test.

## 5. Conclusions

This study is the first to describe the genetic landscape of genomic susceptibility to severe COVID-19 in Slovenia. Several rare genomic variants classified as pathogenic/likely pathogenic were identified as promising candidates related to a higher risk of severe SARS-CoV-2 infection outcome, and represent additional evidence of the importance of such variants in severe COVID-19 infection. While pathogenic variants in *TNFRSF13B* have been shown in individual cases to result in COVID-19 mortality, of the individual variants in the other main genes identified in our study, to the best of our knowledge, none were previously consistently individually associated with mortality due to any viral infection in particular. While the *RNASEL* gene was shown to play a role during Dengue virus infection [35], *MEFV* and *MASP2* deficiencies may manifest subclinically, and *CFTR* deficiency leads to severe outcomes that predominantly involve opportunistic bacterial rather than viral infections. 

This leads us to the conclusion that the identified variants in these genes are not SARS-CoV-2-specific but represent a ‘genetic risk’ background that is very likely shared with susceptibility to many other pathogens. Our results represent an insight into the Slovenian genomic diversity associated with severe COVID-19 outcomes, with possible implications for future public health preventative measures.

## Figures and Tables

**Figure 1 ijms-25-07674-f001:**
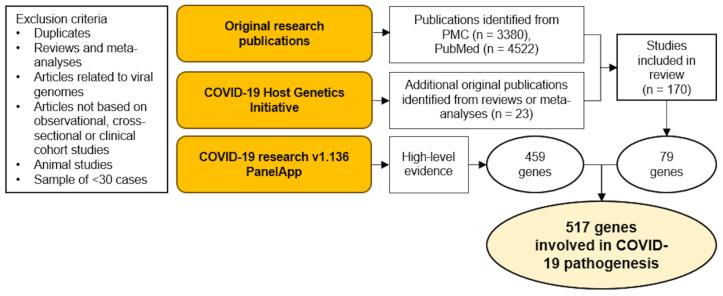
Evidence-based gene panel design.

**Figure 2 ijms-25-07674-f002:**
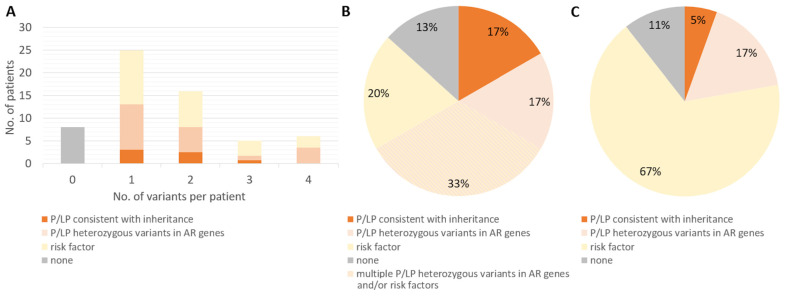
Distribution of pathogenic variants, likely pathogenic variants, and risk factors in COVID-19-associated genes among 60 patients and population controls. (**A**) Number of patients with variants of interest. (**B**) Prevalence and classification of 36 variants of interest in 60 patients. Of the 10 (17%) patients with P/LP variants consistent with inheritance, 6 (10%) had additional variants of interest. In total, multiple variants of interest were identified in 27 of 60 patients (45%). (**C**) Prevalence and classification of 36 variants of interest in 8025 control population individuals. P/LP = pathogenic/likely pathogenic.

**Table 1 ijms-25-07674-t001:** Pathogenic variants, likely pathogenic variants, and risk factors in COVID-19-associated genes.

Gene	Transcript	Variant	Protein Change	No. of Patients ^a^	Patient Allelic Freq.	SGDB Allelic Freq.	*p*-Value (Allelic Freq) ^b^	*p*-Value Adjusted (Allelic Freq) ^d^	ACMG Classification
*ADAR*	NM_001365045.1	c.604C>G	p.Pro202Ala	1	0.0083	0.0019	0.211	0.329	P
*AIRE*	NM_000383.4	c.769C>T	p.Arg257*	2	0.0169	0.0033	0.062	0.152	P
** *APOE* **	NM_000041.4	**c.388T>C**	**p.Cys130Arg**	7	0.0667	0.1059	0.164 ^c^	0.280	R
*ATM*	NM_000051.3	c.6095G>A	p.Arg2032Lys	1	0.0083	0.0003	0.039	0.120	LP
*BRCA2*	NM_000059.3	c.7806-2A>G	splice variant	1	0.0083	0.0007	0.091	0.196	P
*C2*	NM_001282459.2.7	c.841_868del	p.Val281fs	1	0.0085	0.0114	1	1	P
*C6*	NM_000065.4	c.2381+2T>C	splice variant	1	0.0083	0.0054	0.480	0.590	P
*C7*	NM_000587.4	c.1561C>A	p.Arg521Ser	1	0.0083	0.0082	0.627	0.731	P
*C7*	NM_000587.4	c.1924_1925del	p.His643fs	1	0.0083	0.0003	0.046	0.134	P
*C8B*	NM_000066.4	c.1282C>T	p.Arg428*	1	0.0083	0.0072	0.579	0.679	P
*C9*	NM_001737.5	c.162C>A	p.Cys54*	1	0.0083	0.0061	0.522	0.630	P
*CFD*	NM_001317335.2	c.286delG	p.Glu96fs	1	0.0083	0.0006	0.076	0.177	LP
*CFI*	NM_001318057.2	c.111dupA	p.Tyr38fs	1	0.0083	0.0011	0.133	0.246	LP
*CFTR*	NM_000492.4	c.1210-11T>G	intronic	1	0.0089	0.0107	1	1	P
** *CFTR* **	**NM_000492.4**	**c.1727G>C**	**p.Gly576Ala**	1 (CH)	0.0083	0.0038	0.370	0.482	LP
** *CFTR* **	**NM_000492.4**	**c.2002C>T**	**p.Arg668Cys**	1 (CH)	0.0083	0.0058	0.503	0.613	LP
*CFTR*	NM_000492.4	c.2991G>C	p.Leu997Phe	1	0.0088	0.0026	0.262	0.380	LP
*CFTR*	NM_000492.4	c.3154T>G	p.Phe1052Val	1	0.0083	0.0008	0.098	0.205	P
*CFTR*	NM_000492.4	c.3485G>T	p.Arg1162Leu	2	0.0167	0.0070	0.211	0.329	LP
*CYBA*	NM_000101.4	c.222delC	p.Ala75fs	1	0.0083	0	0.142	0.255	LP
*FANCC*	NM_000136.3	c.487_490del	p.Glu163fs	1	0.0083	0	0.142	0.255	LP
*HAVCR2*	NM_032782.5	c.291A>G	p.Ile97Met	4	0.0333	0.0179	0.171	0.287	LP
*HPS4*	NM_001349900.2	c.649C>T	p.Arg217*	1	0.0083	0.0011	0.133	0.246	P
*IL36RN*	NM_012275.3	c.338C>T	p.Ser113Leu	1	0.0083	0.0039	0.379	0.488	P
** *MASP2* **	**NM_006610.4**	**c.359A>G**	**p.Asp120Gly**	8 (1 hom)	0.0750	0.0490	0.189 ^c^	0.308	LP
** *MBL2* **	**NM_000242.2**	**c.154C>T**	**p.Arg52Cys**	13	0.1083	0.0724	0.131 ^c^	0.246	R
** *MBL2* **	**NM_000242.2**	**c.161G>A**	**p.Gly54Asp**	21	0.1917	0.1422	0.123 ^c^	0.233	R
** *MEFV* **	**NM_000243.2**	**c.2084A>G**	**p.Lys695Arg**	5	0.0417	0.0266	0.256	0.377	LP
** *MEFV* **	**NM_000243.2**	**c.2230G>T**	**p.Ala744Ser**	1	0.0083	0.0036	0.353	0.468	LP
*PGM3*	NM_001199917.2	c.463delA	p.Arg155fs	1	0.0083	0.0002	0.041	0.124	LP
*POLR3A*	NM_007055.4	c.1771-7C>G	intronic	1	0.0083	0.0001	0.023	0.090	P
** *PRF1* **	**NM_001083116.3**	**c.272C>T**	**p.Ala91Val**	7	0.0583	0.0564	0.928	1	R
** *RNASEL* **	**NM_021133.4**	**c.1567-11_1574del**	**p.Asp523fs**	1	0.0083	0.0005	0.061	0.152	LP
*RNU4ATAC*	NR_023343.1	n.8C>T	ncRNA	1	0.0083	0.0015	0.180	0.299	LP
*RNU4ATAC*	NR_023343.1	n.40C>T	ncRNA	1	0.0083	0.0005	0.082	0.187	P
** *TNFRSF13B* **	**NM_012452.3**	**c.310T>C**	**p.Cys104Arg**	1	0.0083	0.0029	0.300	0.412	P
*(LZTFL1)* rs17713054	NA	rs17713054-A	intergenic	14	0.1250	0.1055	0.522 ^c^	NA	RL
*(LZTFL1)* rs35482426	NM_001276378.2	c.-137-18021_-137-18020del	intronic	15	0.1333	0.1441	0.753 ^c^	NA	RL

LP = likely pathogenic; P = pathogenic; R = risk factor; RL = risk locus; CH = in compound heterozygous state; hom = in homozygous state; SGDB = Slovenian genomic database. Variants for which the genotype was consistent with the proposed inheritance model for the gene are indicated in **bold.** ^a^ In heterozygous state, if not indicated otherwise. ^b^ Fisher’s exact test for comparison between patients’ and SGBD allelic frequency or ^c^ χ^2^ test if expected frequency was ≥5. ^d^
*p* values adjusted for 689 variants in 517 genes.

**Table 2 ijms-25-07674-t002:** Distribution of pathogenic and likely pathogenic variants, and risk factors in COVID-19-associated genes among 60 patients.

		Patients																																																												
Variant	Class	1	2	3	4	5	6	7	8	9	10	11	12	13	14	15	16	17	18	19	20	21	22	23	24	25	26	27	28	29	30	31	32	33	34	35	36	37	38	39	40	41	42	43	44	45	46	47	48	49	50	51	52	53	54	55	56	57	58	59	60	SUM
*ADAR*:c.604C>G	P																																			1																										1
*AIRE*:c.769C>T	P	1																																																									1			2
***APOE*:c.388T>C**	R					1		1		1									1								1																		1^hom^														1			7
*ATM*:c.6095G>A	LP																				1																																									1
*BRCA2*:c.7806-2A>G	P		1																																																											1
*C2*:c.841_868del	P	1																																																												1
*C6*:c.2381+2T>C	P														1																																															1
*C7*:c.1561C>A	P														1																																															1
*C7*:c.1924_1925del	P																																											1																		1
*C8B*:c.1282C>T	P																																			1																										1
*C9:*c.162C>A	P																												1																																	1
*CFD*:c.286del	LP																																																								1					1
*CFI*:c.111dup	LP																								1																																					1
*CFTR*:c.1210-11T>G	P																																												1																	1
***CFTR*:c.1727G>C**	LP						1																																																							1
***CFTR*:c.2002C>T**	LP						1																																																							1
*CFTR*:c.2991G>C	LP																																																1													1
*CFTR*:c.3154T>G	P																																							1																						1
*CFTR*:c.3485G>T	LP																																													1									1							2
*CYBA*:c.222del	LP																																1																													1
*FANCC*:c.487_490del	LP																																																	1												1
*HAVCR2*:c.291A>G	LP			1								1																									1																					1				4
*HPS4*:c.649C>T	P																																																					1								1
*IL36RN*:c.338C>T	P																					1																																								1
***MASP2*:c.359A>G hom**	LP																																									1																				1
*MASP2*:c.359A>G het	LP	1											1																1			1				1																	1				1					7
***MBL2*:c.154C>T**	R	1			1	1																			1						1	1				1	1					1			1				1								1		1			13
***MBL2*:c.161G>A**	R					1		1	1	1	1			1^hom^		1			1	1				1					1	1				1					1	1								1		1	1	1	1^hom^				1					21
***MEFV*:c.2084A>G**	LP			1														1					1												1						1																					5
***MEFV*:c.2230G>T**	LP																													1																																1
*PGM3*:c.463del	LP																															1																														1
*POLR3A*:c.1771-7C>G	P																																																1													1
***PRF1*:c.272C>T**	R								1														1						1	1															1						1									1		7
***RNASEL*:c.1567-11_1574del**	LP								1																																																					1
*RNU4ATAC*:n.40C>T	P																															1																														1
*RNU4ATAC*:n.8C>T	LP																														1																															1
***TNFRSF13B* c.310T>C**	P																																1																													1
*LZTFL1* rs17713054	RL			1							1	1	1				1		1								1			1										1	1		1						1				1		1							14
*LZTFL1* rs35482426	RL			1				1			1	1	1				1		1								1			1				1						1	1		1										1		1							15
**SUM**		**4**	**1**	**2**	**1**	**3**	**2**	**2**	**3**	**2**	**1**	**1**	**1**	**1**	**2**	**1**	**0**	**1**	**2**	**1**	**1**	**1**	**2**	**1**	**2**	**0**	**1**	**0**	**4**	**3**	**2**	**4**	**2**	**1**	**1**	**4**	**2**	**0**	**1**	**2**	**1**	**2**	**0**	**1**	**4**	**1**	**0**	**1**	**3**	**2**	**2**	**1**	**2**	**1**	**1**	**0**	**4**	**1**	**3**	**1**	**0**	**96**

Deceased patients are underlined. Variants for which the genotype was consistent with the proposed inheritance model for the gene are indicated in **bold**. LP = likely pathogenic (red); P = pathogenic (red); R = risk factor (yellow); RL = risk locus; hom = homozygous; het = heterozygous.

**Table 3 ijms-25-07674-t003:** Demographic and clinical data of included hospitalized patients.

	Total
Subjects n or n (%)	60
Age mean ± SD years	77 ± 9.9
Male sex n (%)	31 (52)
**Previous coexisting disease n (%)**	
Type 2 diabetes	17 (28)
Heart disease *	28 (47)
Hypertension	38 (63
Chronic lung disease	16 (27)
Rheumatic diseases	5 (8)
Cancer	10 (17)
Chronic kidney disease	7 (12)
**Number of coexisting diseases n (%)**	
None	4 (7)
One	18 (30)
Two or more	38 (63)
Died (%)	7 (12)

***** Excluding hypertension.

## Data Availability

The data presented in this study are available on request from the corresponding author. The raw data are not publicly available due to containing additional potentially sensitive genomic information.

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
