# Peer review of "Genomic Landscape of Susceptibility to Severe COVID-19 in the Slovenian Population"

_ijms, 2024, doi:10.3390/ijms25147674_

Round 1

Reviewer 1 Report

Comments and Suggestions for Authors

The SARS-CoV-2 virus, responsible for the recent Covid-19 pandemic that took millions of lives across the world, is nonetheless noteworthy for causing a wide range of disease manifestations.  Infected individuals can present with very mild, almost asymptomatic upper respiratory infections, moderate disease or extremely severe disease with high fever, cough, extreme fatigue, diarrhea, acute respiratory symptoms, including pneumonia, and cardiac symptoms.  A concerningly high number of infected patients required hospitalization and approximately 3.4%, unfortunately succumbed to the disease. 

There is a well-characterized association between severe Covid-19 disease and morbidity and underlying complications, such as respiratory and cardiac disease, diabetes, obesity, etc.  On the virus side, being an RNA virus, SARS-CoV-2 exists as a continually evolving set of variants with different clinical manifestations.  However, all of these associations do not account entirely for Covid-19 susceptibility in the population.  Certainly, age plays a major role in disease severity.  However, there is likely to be an, as yet, uncharacterized relationship between Covid-19 disease severity and the genetic make-up of infected individuals.

In this excellent study, an effort is mounted to begin to understand the contribution of genetics to susceptibility to severe Covid-19 disease.  Although early studies identified a tantalizing association between blood groups and severity of infection, subsequent studies have implicated several hundred genes as having a role in the genetic contribution to severe Covid-19 with many common genomic variants standing out in severe Covid-19 patients.  To address this issue further, the group sequenced the full genomes of 60 patients who had been hospitalized with severe Covid-19 in Slovenia.  The goal was to attempt to identify variants of interest in 517 genes previously implicated in susceptibility to severe Covid-19.  This effort identified pathogenic, or at least likely pathogenic variants, in 52 of the 60 patients, 25 of which had only a single variant and 27 of which had multiple variants.

While the complexity of the relationship between severe Covid-19 and host genetics has become apparent, this group identified a total of 7 pathogenic/likely pathogenic variants in a total of 10 patients.  These were mapped to 5 genes, CFTR, MASP2, MEFV, RNASEL and TNFRSF13B, which, in many instances, act in concert.  Perhaps not surprisingly, all of these genes are associated with immunodeficiency, susceptibility to infection and inflammatory disorders, which is, of course, consistent with their impact on disease severity.

This manuscript details the findings from an exceptionally well-designed and appropriately interpreted study that constitutes an important contribution to our understanding of the relationship between the human genome and Covid-19 disease severity.  There are no detected weaknesses in any aspect of the study or its conclusions.  These findings will likely provide a road map and serve as the starting point for future studies on the topic.  Its impact on the field is considered high.

Author Response

We thank the Reviewer 1 for their positive comments. We attest we have revised the manuscript according to all suggestions by all four Reviewers. Please find the final version attached in the system.

Reviewer 2 Report

Comments and Suggestions for Authors

Good morning to all the authors,

Analyzing the manuscript with ID ijms-3065851-peer-review-v1, entitled "Genomic landscape of susceptibility to severe COVID-19 in the Slovenian population" for possible publication in the Journal IJMS,

I consider that:

1. The authors proposed a much-discussed topic in today's medical scientific world: the factors that influence the occurrence of SARS-CoV-2 infection.

2. This article follows all the specific instructions of the journal presented in aims and scope, instructions for authors, and other information about the Journal IJMM.

3. In Chapter Introduction: The authors present relevant and important medical information for the chosen topic: the coronavirus pandemic.

4. In Chapter Results: The results of this study are well presented and easy to understand in the text of the manuscript. Tables 1, 2, Supplementary 2, and Figure 1 (A, B, C) are suggestive of the study conducted and very clearly presented!

5. In Chapter Discussion: The authors compared all the results obtained in this study with the results of other studies and the assessments of other authors, according to the bibliography.

6. In Chapter Limitations: The authors have clearly presented all the limitations of their study, thus proving a high professional ethics!

7. In Chapter Conclusions: The authors clearly presented the conclusions of the study.

8. In Chapter Materials and Methods: The authors made a special effort from the beginning of the study and presented very clearly in Table 1 supplementary and Figure 2.

- The number of investigated patients is relevant for this study.

- They had the inclusion and exclusion criteria for the selected patients. Table 3 is relevant to this study.

- They had the patients' informed consent.

- They had the approval of the Ethics Committee of the hospital.

- For the genetic investigations (of the genome), the authors used technologies, standardized work protocols, high-performance medical equipment, and certified kits/panels. This is the main reason for the results obtained are clear! Supplementary Table 3 is relevant to the results obtained in the study.

- For the statistical analysis of the data, the authors used high-performance software: software version 4.1.3.

9. The bibliography chosen by the authors corresponds to the requirements and refers to the subject of the article.

10. The authors obtained funding from the Government of their country for this study. This demonstrates the high scientific performance of the study!

11.  All authors have made an equitable contribution to the study.

In conclusion:

I ACCEPT for the publication of this article in the International Journal of Molecular Sciences!

Congratulations to all authors for this article!

Čestitke vsem avtorjem za ta članek! 

Author Response

(The authors gave the same response as above.)

Reviewer 3 Report

Comments and Suggestions for Authors

In the paper by Kovanda et al, entitled "Genomic landscape of susceptibility to severe COVID-19 in the Slovenian population", they identify rare genomic variants that may be associated with an increased risk of severe SARS-CoV-2 infection. The authors presented a well-conducted study that could be improved. They provided information to answer the following questions:

How did the extensive literature review, in which the authors found 517 genes associated with severe SARS-CoV-2 infection, contribute to the results of the whole genome sequencing?

What is the expressivity and penetrance of the genes MEFV, CFTR, RNASEL, TNFRSF13B and MASP2 found in the Slovenian population among the 517 genes reported in the literature?

In what other viral infections are the variants found in this paper, such as MEFV, CFTR, RNASEL, TNFRSF13B, and MASP2, associated with mortality?

What emergent dominant SARS-CoV-2 variants of concern were present in the region at the time the study samples were collected?

The authors must include in their conclusions the results of their extensive literature review, in which the authors found 517 genes associated with severe SARS-CoV-2 infection.

What other inclusion criteria were used in the interventions in Supplementary Data S1?

Author Response

Comments 1: In the paper by Kovanda et al, entitled "Genomic landscape of susceptibility to severe COVID-19 in the Slovenian population", they identify rare genomic variants that may be associated with an increased risk of severe SARS-CoV-2 infection. The authors presented a well-conducted study that could be improved. They provided information to answer the following questions:

Response 1: We thank the Reviewer 3 for the positive comment.

Comments 2: How did the extensive literature review, in which the authors found 517 genes associated with severe SARS-CoV-2 infection, contribute to the results of the whole genome sequencing?

Response 2: We would like to clarify that upon starting the study, not much was known in terms of human genetics as relating to SARS-CoV-2 infection outcome. Therefore, initially we aimed to look at all of the scientific evidence and curate our own gene panel. However, during the course of the study results from large international initiatives became available and therefore we chose to integrate the evidence obtained through literature search, with this high-quality additional evidence. It was in this way that we achieved the final list of 517 genes.

Comments 3: What is the expressivity and penetrance of the genes MEFV, CFTR, RNASEL, TNFRSF13B and MASP2 found in the Slovenian population among the 517 genes reported in the literature?

Response 3: The main expression of MEFV occurs in mature granulocytes, with an alternative isoform expressed in leukocytes. The penetrance of MEFV is complex, ranging from asymptomatic heterozygotes, to heterozygotes with clinical manifestation of Mediterranean fever. In the Slovenian population, the identified K695R variant is relatively common in apparently healthy individuals as evident from its SGDB prevalence and previous studies examining the carrier rate in Slovenia [1]. Similarly, MASP2 expressivity, measured by its plasma level, varies in different populations [2] and its penetrance is highly variable, with estimated 5-18% of MASP2 deficient individuals remaining asymptomatic [2,3]. The penetrance of CFTR is similarly complex, ranging from autosomal dominant in case of bronchiestatis (OMIM:211400) to autosomal recessive in case of cystic fibrosis, with expressivity being variant dependent. The heterozygous, homozygous and compound heterozygous variants in TNFRSF13B lead to common variable immunodeficiency or selective IgA deficiency (OMIM: 240500), that are characterized by susceptibility to recurrent infections. Finally, RNASEL is involved in intracellular single RNA cleavage as part of the dsRNA detection system and antiviral response [4], but also, the dysfunction of RNASEL increases the risk for cancer and autoimmune disease [5] in an apparently autosomal dominant manner. The allelic frequency of the variants in MEFV, CFTR, RNASEL, TNFRSF13B and MASP2 in the Slovenian population, based on >8000 individuals, are given in Table 1 column ‘SGDB allelic frequency’. While MASP2 and MEFV are relatively common, CFTR, RNASEL and TNFRSF13 are rare. In the text SGDB is explained as the Slovenian Genomic Database, but to make this clearer we have also included the abbreviation in the figure legend.

Comments 4: In what other viral infections are the variants found in this paper, such as MEFV, CFTR, RNASEL, TNFRSF13B, and MASP2, associated with mortality?

Response 4: Of the individual variants in the main genes identified in our study, to the best of our knowledge, none were consistently individually associated with mortality due to any viral infection in particular. While RNASEL has shown to play a role during Dengue virus infection [6], MEFV and MASP2 deficiencies may manifest subclinically, and CFTR deficiency leads to a severe outcome but predominantly opportunistic bacterial infections are involved. However, as we have pointed out in the discussion, heterozygous pathogenic variants in CFTR have been more frequent in hospitalized COVID-19 patients [7].

Similarly, TNFRSF13B deficiency is associated with common variable immunodeficiency (CVID) that is characterized by susceptibility for severe infection with many different infectious agents [8]. While CVID has been linked to a fatal pediatric case of COVID-19 [9], this proband had an additional genetic finding in TBK1, also linked to immune dysregulation.

Therefore, our conclusion would be that the identified variants in these genes are not SARS-CoV-2 specific but represent a ‘genetic risk’ background that is very likely shared with susceptibility to many other pathogens.

Comments 5: What emergent dominant SARS-CoV-2 variants of concern were present in the region at the time the study samples were collected?

Response 5: The samples were collected during the second pandemic wave in Slovenia (September-December 2020), and unfortunately we do not have original data on the strain that had infected the participants in this study. In Slovenia, the data collected for the purpose of surveillance of viral variants was collected since January 2021 [10] https://doi.org/10.2807/1560-7917.ES.2023.28.8.2200451. Based on this data from January 2021 data from the country surveillance, an assumption can be made that the B.1 sublineage was dominant at that time.

Comments 6: The authors must include in their conclusions the results of their extensive literature review, in which the authors found 517 genes associated with severe SARS-CoV-2 infection.

Response 6: We thank the reviewer for this suggestion. To make the manuscript clearer we have added the following sentence to the Results:

‘We sequenced the whole genomes of 60 patients, hospitalized during the second pandemic wave of COVID-19 in Slovenia [11] to identify variants of interest in 517 evidence-based genes associated with severe COVID-19. This evidence-based gene panel was constructed by combining three strategies: literature curation of original research publications; high-impact targets identified by the GWAS COVID-19 Host Genetic Initiative (COVID-19 HGI) [12]; and high-support genes from a research COVID-19 associated gene panel (COVID-19 research v1.136 PanelApp), curated by EnglandGenomics [13] (Figure 2). The identified variants were classified according to ACMG Criteria [14] and the model of inheritance. The resulting pathogenic/likely pathogenic variants and risk factors, as well as their Slovenian genomic database (SGDB) background prevalence, are given in Table 1.’

As shown in corrected Figure 2 we included 170 studies resulting in 79 genes in the final 517 panel, of which most were also contained in the 459 genes with high-level evidence from the COVID-19 research v1.136 PanelApp. As shown in the methods, because of the sheer volume of publications pertaining to COVID-19 research after 09.05.2022 and the availability of pre-curated data from the HGI initiative and PanelApp, the candidate genes were aggregated to the final 517. We have now included the publication results as an additional tab in the final data curation in Supplementary Data S2.

Comments 7: What other inclusion criteria were used in the interventions in Supplementary Data S1?

Response 7: As we write in the methods we would like to explain that when screening abstracts of the >3000 publication we were faced with highly variable wording pertaining to sequencing. In the PICO table we state some, such as WES that can also be worded as ‘whole exome sequencing’ or ‘whole-exome-sequencing’, and of course clinical exome sequencing (CES), exome sequencing (ES), targeted-sequencing, panel-sequencing, etc. are acceptable. The interventions inclusion criteria were therefore wide and each publication was individually assessed for inclusion of genetic testing, to exclude studies lacking genotyping data. The following search strings were used to identify relevant works: (<((genetic variation*) OR (gene polymorphism*) OR (polymorphism*) OR (gene variant*) OR (allele*) OR(GWAS*) OR (variant*) OR (host genetics*) OR (genetic susceptibility*) OR (genome*wide association study*)) AND ( (COVID-19*) OR (SARS-CoV-2*) OR (coronavirus*)) > [[Title/Abstract]]). As the Reviewer has suggested, we have included this additional information in the Supplementary table 1.

PICO

Inclusion criteria

Exclusion criteria

Interventions

Genotyping, sequencing, GWAS (genome wide association study), WES (whole exome sequencing), inclusive of any suitable method for the detection of genetic variation, gene polymorphism, gene variant, allele(s), variant(s), host genetic(s), genetic susceptibility.

No genotype data

Additional references included in the revised manuscript

  1. Debeljak, M.; Abazi, N.; Toplak, N.; Stavrić, K.; Kolnik, M.; Kuzmanovska, D.; Avčin, T. Prevalence of MEFV Gene Mutations in Apparently Healthy Slovenian and Macedonian Population. Pediatr. Rheumatol. 2011, 9, P301, 1546-0096-9-S1-P301, doi:10.1186/1546-0096-9-S1-P301.
  2. Thiel, S.; Steffensen, R.; Christensen, I.J.; Ip, W.K.; Lau, Y.L.; Reason, I.J.M.; Eiberg, H.; Gadjeva, M.; Ruseva, M.; Jensenius, J.C. Deficiency of Mannan-Binding Lectin Associated Serine Protease-2 Due to Missense Polymorphisms. Genes Immun. 2007, 8, 154–163, doi:10.1038/sj.gene.6364373.
  3. Sokolowska, A.; Szala, A.; St Swierzko, A.; Kozinska, M.; Niemiec, T.; Blachnio, M.; Augustynowicz-Kopec, E.; Dziadek, J.; Cedzynski, M. Mannan-Binding Lectin-Associated Serine Protease-2 (MASP-2) Deficiency in Two Patients with Pulmonary Tuberculosis and One Healthy Control. Cell. Mol. Immunol. 2015, 12, 119–121, doi:10.1038/cmi.2014.19.
  4. Silverman, R.H. Viral Encounters with 2′,5′-Oligoadenylate Synthetase and RNase L during the Interferon Antiviral Response. J. Virol. 2007, 81, 12720–12729, doi:10.1128/JVI.01471-07.
  5. Casey, G.; Neville, P.J.; Plummer, S.J.; Xiang, Y.; Krumroy, L.M.; Klein, E.A.; Catalona, W.J.; Nupponen, N.; Carpten, J.D.; Trent, J.M.; et al. RNASEL Arg462Gln Variant Is Implicated in up to 13% of Prostate Cancer Cases. Nat. Genet. 2002, 32, 581–583, doi:10.1038/ng1021.
  6. Lin, R.-J.; Yu, H.-P.; Chang, B.-L.; Tang, W.-C.; Liao, C.-L.; Lin, Y.-L. Distinct Antiviral Roles for Human 2’,5’-Oligoadenylate Synthetase Family Members against Dengue Virus Infection. J. Immunol. Baltim. Md 1950 2009, 183, 8035–8043, doi:10.4049/jimmunol.0902728.
  7. Baldassarri, M.; Fava, F.; Fallerini, C.; Daga, S.; Benetti, E.; Zguro, K.; Amitrano, S.; Valentino, F.; Doddato, G.; Giliberti, A.; et al. Severe COVID-19 in Hospitalized Carriers of Single CFTR Pathogenic Variants. J. Pers. Med. 2021, 11, 558, doi:10.3390/jpm11060558.
  8. Resnick, E.S.; Moshier, E.L.; Godbold, J.H.; Cunningham-Rundles, C. Morbidity and Mortality in Common Variable Immune Deficiency over 4 Decades. Blood 2012, 119, 1650–1657, doi:10.1182/blood-2011-09-377945.
  9. Schmidt, A.; Peters, S.; Knaus, A.; Sabir, H.; Hamsen, F.; Maj, C.; Fazaal, J.; Sivalingam, S.; Savchenko, O.; Mantri, A.; et al. TBK1 and TNFRSF13B Mutations and an Autoinflammatory Disease in a Child with Lethal COVID-19. NPJ Genomic Med. 2021, 6, 55, doi:10.1038/s41525-021-00220-w.
  10. Janezic, S.; Mahnic, A.; Kuhar, U.; Kovač, J.; Jenko Bizjan, B.; Koritnik, T.; Tesovnik, T.; Šket, R.; Krapež, U.; Slavec, B.; et al. SARS-CoV-2 Molecular Epidemiology in Slovenia, January to September 2021. Eurosurveillance 2023, 28, doi:10.2807/1560-7917.ES.2023.28.8.2200451.
  11. Šelb, J.; Bitežnik, B.; Bidovec Stojković, U.; Rituper, B.; Osolnik, K.; Kopač, P.; Svetina, P.; Cerk Porenta, K.; Šifrer, F.; Lorber, P.; et al. Immunophenotypes of Anti-SARS-CoV-2 Responses Associated with Fatal COVID-19. ERJ Open Res. 2022, 8, 00216–02022, doi:10.1183/23120541.00216-2022.
  12. The COVID-19 Host Genetics Initiative The COVID-19 Host Genetics Initiative, a Global Initiative to Elucidate the Role of Host Genetic Factors in Susceptibility and Severity of the SARS-CoV-2 Virus Pandemic. Eur. J. Hum. Genet. 2020, 28, 715–718, doi:10.1038/s41431-020-0636-6.
  13. Martin, A.R.; Williams, E.; Foulger, R.E.; Leigh, S.; Daugherty, L.C.; Niblock, O.; Leong, I.U.S.; Smith, K.R.; Gerasimenko, O.; Haraldsdottir, E.; et al. PanelApp Crowdsources Expert Knowledge to Establish Consensus Diagnostic Gene Panels. Nat. Genet. 2019, 51, 1560–1565, doi:10.1038/s41588-019-0528-2.
  14. The ACMG Laboratory Quality Assurance Committee; Richards, S.; Aziz, N.; Bale, S.; Bick, D.; Das, S.; Gastier-Foster, J.; Grody, W.W.; Hegde, M.; Lyon, E.; et al. Standards and Guidelines for the Interpretation of Sequence Variants: A Joint Consensus Recommendation of the American College of Medical Genetics and Genomics and the Association for Molecular Pathology. Genet. Med. 2015, 17, 405–423, doi:10.1038/gim.2015.30.
  15. Golinelli, D.; Boetto, E.; Maietti, E.; Fantini, M.P. The Association between ABO Blood Group and SARS-CoV-2 Infection: A Meta-Analysis. PloS One 2020, 15, e0239508, doi:10.1371/journal.pone.0239508.
  16. Liu, N.; Zhang, T.; Ma, L.; Zhang, H.; Wang, H.; Wei, W.; Pei, H.; Li, H. The Impact of ABO Blood Group on COVID-19 Infection Risk and Mortality: A Systematic Review and Meta-Analysis. Blood Rev. 2020, 100785, doi:10.1016/j.blre.2020.100785.
  17. Di Maria, E.; Latini, A.; Borgiani, P.; Novelli, G. Genetic Variants of the Human Host Influencing the Coronavirus-Associated Phenotypes (SARS, MERS and COVID-19): Rapid Systematic Review and Field Synopsis. Hum. Genomics 2020, 14, 30, doi:10.1186/s40246-020-00280-6.
  18. Elhabyan, A.; Elyaacoub, S.; Sanad, E.; Abukhadra, A.; Elhabyan, A.; Dinu, V. The Role of Host Genetics in Susceptibility to Severe Viral Infections in Humans and Insights into Host Genetics of Severe COVID-19: A Systematic Review. Virus Res 2020, doi:10.1016/j.virusres.2020.198163.
  19. Ramos-Lopez, O.; Daimiel, L.; Ramírez de Molina, A.; Martínez-Urbistondo, D.; Vargas, J.A.; Martínez, J.A. Exploring Host Genetic Polymorphisms Involved in SARS-CoV Infection Outcomes: Implications for Personalized Medicine in COVID-19. Int. J. Genomics 2020, 2020, 6901217, doi:10.1155/2020/6901217.
  20. Pairo-Castineira, E.; Rawlik, K.; Bretherick, A.D.; Qi, T.; Wu, Y.; Nassiri, I.; McConkey, G.A.; Zechner, M.; Klaric, L.; Griffiths, F.; et al. GWAS and Meta-Analysis Identifies 49 Genetic Variants Underlying Critical COVID-19. Nature 2023, 617, 764–768, doi:10.1038/s41586-023-06034-3.
  21. Severe Covid-19 GWAS Group; Ellinghaus, D.; Degenhardt, F.; Bujanda, L.; Buti, M.; Albillos, A.; Invernizzi, P.; Fernández, J.; Prati, D.; Baselli, G.; et al. Genomewide Association Study of Severe Covid-19 with Respiratory Failure. N. Engl. J. Med. 2020, 383, 1522–1534, doi:10.1056/NEJMoa2020283.
  22. Liu, P.; Fang, M.; Luo, Y.; Zheng, F.; Jin, Y.; Cheng, F.; Zhu, H.; Jin, X. Rare Variants in Inborn Errors of Immunity Genes Associated With Covid-19 Severity. Front. Cell. Infect. Microbiol. 2022, 12, 888582, doi:10.3389/fcimb.2022.888582.
  23. Khadzhieva, M.B.; Gracheva, A.S.; Belopolskaya, O.B.; Kolobkov, D.S.; Kashatnikova, D.A.; Redkin, I.V.; Kuzovlev, A.N.; Grechko, A.V.; Salnikova, L.E. COVID-19 Severity: Does the Genetic Landscape of Rare Variants Matter? Front. Genet. 2023, 14, 1152768, doi:10.3389/fgene.2023.1152768.

Reviewer 4 Report

Comments and Suggestions for Authors

Introduction

The introduction section presents the idea of the work and characterizes the disease well. However, there are few discussions on the practical applications/impact of SNPs on disease manifestation/severity. Please improve this aspect.

Results and Discussion:

The results are clearly presented in a logical sequence.

Regarding Table 2: Is it necessary for the manuscript? I think this table could be removed or inserted into the supplementary section since you do not explore the clinical individual factors together with the variants analysis.

Why do we not see results from the control group? Perhaps complementary analysis showing comparisons like Figure 1 could improve the paper and reduce the focus on the study's limitations.

The discussion could avoid recapitulating the results previously presented and instead focus on constructing a concise process of idea development, leading to a conclusion that is more specific to the data and not generic.

Methods
The methods are adequate and highly related to results presented in the manuscript. Aiming improves the section: a more concise construction of each step for study development.

Minor comments:

In the third line: I suggest changing “critical” to “severe”. (Introduction)

Figure 1 B – the LP/P and AR abbreviations need to be explained in the legend. (Results)

The following paragraphs are unclear: (Discussion)

“Additionally, we identified risk factor variants in 58% (35/60) of our patients, while they were the sole finding in 28% (17/60) of our patient cohort. A total of 4 risk factor variants were identified in the following 3 genes; APOE, MBL2, and PRF1.” This data (28%-17/60) was previously presented? Was not clear to me.

Comments on the Quality of English Language

The manuscript would benefit from a thorough revision of the English language. Some parts of the text are unclear or confusing. A native English speaker could help improve this.

Author Response

Reviewer 4

Introduction

Comments 1: The introduction section presents the idea of the work and characterizes the disease well. However, there are few discussions on the practical applications/impact of SNPs on disease manifestation/severity. Please improve this aspect.

Response1: I order to illustrate how SNP may prove helpful in identifying those most at risk of developing severe disease, we have included the following text in the introduction section:

‘Indeed, because of genetics’ potential importance in disease prognosis, prevention, and public health planning measures, the field of COVID-19-host genetics has expanded rapidly resulting in several thousand publications on this topic in the last four years. Disease-associated population-specific variant information is particularly important as it is prerequisite for any future development of screening tests to identify high-risk individuals.

The results of this research have been more complex. Following early reports on the association between blood groups and occurrence of infection [15,16], and initial examination of a handful of candidate genes, based on their association with other viral interactions, such as ACE2, CLEC4M, MBL, ACE, CD209, FCER2, OAS-1, TLR4, and TNF-α [17–19], in the years since, several hundred genes have been implicated to play a role in the complex host genetic contribution to COVID-19. Genome-wide association studies (GWAS) have shown that many common genomic variants are enriched in cohorts of patients with severe COVID-19 [20,21]. So far, the identified polymorphisms show geographical differences, and were mostly shown to have weak effects and even combining them to assess their polygenic risk score (PRS) so far has not been able to effectively predict disease outcome. Similarly, while it has been proposed that the accumulation of weak effects of many rare functional variants may contribute to the overall risk in patients with severe disease [22,23], the information on such variants is scarce in many populations, including ours.

Therefore, we aimed to determine rare genomic variants and their burden in a comprehensive set of evidence-based genes in well-characterized patients hospitalized due to COVID-19 in Slovenia.

Comments 2: Results and Discussion:

The results are clearly presented in a logical sequence.

Regarding Table 2: Is it necessary for the manuscript? I think this table could be removed or inserted into the supplementary section since you do not explore the clinical individual factors together with the variants analysis.

Response 2: We would like to answer that we have included the table 2 to better illustrate the complexity of simultaneous findings (i.e. more than 1 variant of interest was identified in over half of our patients with variants of interest). The table 2 enables the reader to better understand the explanation given in the results regarding the variants of interest, heterozygosity considerations, and risk-factors that do not constitute coding variants, such as LZTFL1 rs17713054 and rs35482426, and that are discussed independently of the variants of interest (please see our comment on the risk-factors below). Of course, if the Reviewer and Editor insist, we are prepared to move the Table 2 to the Supplementary section of the manuscript.

Comments 3: Why do we not see results from the control group? Perhaps complementary analysis showing comparisons like Figure 1 could improve the paper and reduce the focus on the study's limitations.

Response 3: We would like to explain that the control group comparison (allelic frequency) is given in Table 1; Table 1, column ‘SGDB allelic frequency’. In the text SGDB is explained as the Slovenian Genomic Database, but to make this clearer we have now also included the abbreviation in the figure legend. Because the SGDB database contains summary population data and is de-identified, it is unfortunately not possible to see co-occuring variants in single patients (ie. It is not possible to construct something like table 2 for 8025 control individuals from SGDB).

Comments 4: The discussion could avoid recapitulating the results previously presented and instead focus on constructing a concise process of idea development, leading to a conclusion that is more specific to the data and not generic.

Response 4: We thank the Reviewer for this suggestion. In order to illustrate our ideas more clearly, we have extensively re-written parts of the discussion (marked in red in the manuscript). Additional references are included at the bottom of this reply. Please note that our study, similarly to others had specific limitations, that prevented us from making more specific conclusions. Nevertheless, its merit remains in that it adds evidence on the importance of rare deleterious variants present in our population, on the progression of the SARS-CoV-2 infection.

Comments 5: Methods
The methods are adequate and highly related to results presented in the manuscript. Aiming improves the section: a more concise construction of each step for study development.

Minor comments:

In the third line: I suggest changing “critical” to “severe”. (Introduction)

Response 5: We have corrected the sentence in question.

Comments 6: Figure 1 B – the LP/P and AR abbreviations need to be explained in the legend. (Results)

Response 6: We have corrected the sentence in question.

Comments 7: The following paragraphs are unclear: (Discussion)

“Additionally, we identified risk factor variants in 58% (35/60) of our patients, while they were the sole finding in 28% (17/60) of our patient cohort. A total of 4 risk factor variants were identified in the following 3 genes; APOE, MBL2, and PRF1.” This data (28%-17/60) was previously presented? Was not clear to me.

Response 7: We would like to explain that this paragraph pertains to risk factor variants only, explaining their occurrence independently of the other variants of interest. The individual risk factors and their co-occurrences with other variants are shown in Table 2, and the discussion sentence refers to the following sentence in results:

 ‘35 of the 60 patients carried more than 48 risk factor variants in MBL2 (34), PRF1 (7), and APOE (7)(Table 2). The LZTFL1 risk factors rs17713054 and rs35482426 were present in a total of 16 patients, and were the only finding in two patients (Table 2).‘

In order to make this clear, we have included another reference to Table 2 next to the first mention of risk factors in results.

Comments 8: Comments on the Quality of English Language

The manuscript would benefit from a thorough revision of the English language. Some parts of the text are unclear or confusing. A native English speaker could help improve this.

Response 8: We confirm we have re-checked the grammar by a professional English translator and have made relevant grammar corrections in the final manuscript.

Additional references included in the revised manuscript

  1. Debeljak, M.; Abazi, N.; Toplak, N.; Stavrić, K.; Kolnik, M.; Kuzmanovska, D.; Avčin, T. Prevalence of MEFV Gene Mutations in Apparently Healthy Slovenian and Macedonian Population. Pediatr. Rheumatol. 2011, 9, P301, 1546-0096-9-S1-P301, doi:10.1186/1546-0096-9-S1-P301.
  2. Thiel, S.; Steffensen, R.; Christensen, I.J.; Ip, W.K.; Lau, Y.L.; Reason, I.J.M.; Eiberg, H.; Gadjeva, M.; Ruseva, M.; Jensenius, J.C. Deficiency of Mannan-Binding Lectin Associated Serine Protease-2 Due to Missense Polymorphisms. Genes Immun. 2007, 8, 154–163, doi:10.1038/sj.gene.6364373.
  3. Sokolowska, A.; Szala, A.; St Swierzko, A.; Kozinska, M.; Niemiec, T.; Blachnio, M.; Augustynowicz-Kopec, E.; Dziadek, J.; Cedzynski, M. Mannan-Binding Lectin-Associated Serine Protease-2 (MASP-2) Deficiency in Two Patients with Pulmonary Tuberculosis and One Healthy Control. Cell. Mol. Immunol. 2015, 12, 119–121, doi:10.1038/cmi.2014.19.
  4. Silverman, R.H. Viral Encounters with 2′,5′-Oligoadenylate Synthetase and RNase L during the Interferon Antiviral Response. J. Virol. 2007, 81, 12720–12729, doi:10.1128/JVI.01471-07.
  5. Casey, G.; Neville, P.J.; Plummer, S.J.; Xiang, Y.; Krumroy, L.M.; Klein, E.A.; Catalona, W.J.; Nupponen, N.; Carpten, J.D.; Trent, J.M.; et al. RNASEL Arg462Gln Variant Is Implicated in up to 13% of Prostate Cancer Cases. Nat. Genet. 2002, 32, 581–583, doi:10.1038/ng1021.
  6. Lin, R.-J.; Yu, H.-P.; Chang, B.-L.; Tang, W.-C.; Liao, C.-L.; Lin, Y.-L. Distinct Antiviral Roles for Human 2’,5’-Oligoadenylate Synthetase Family Members against Dengue Virus Infection. J. Immunol. Baltim. Md 1950 2009, 183, 8035–8043, doi:10.4049/jimmunol.0902728.
  7. Baldassarri, M.; Fava, F.; Fallerini, C.; Daga, S.; Benetti, E.; Zguro, K.; Amitrano, S.; Valentino, F.; Doddato, G.; Giliberti, A.; et al. Severe COVID-19 in Hospitalized Carriers of Single CFTR Pathogenic Variants. J. Pers. Med. 2021, 11, 558, doi:10.3390/jpm11060558.
  8. Resnick, E.S.; Moshier, E.L.; Godbold, J.H.; Cunningham-Rundles, C. Morbidity and Mortality in Common Variable Immune Deficiency over 4 Decades. Blood 2012, 119, 1650–1657, doi:10.1182/blood-2011-09-377945.
  9. Schmidt, A.; Peters, S.; Knaus, A.; Sabir, H.; Hamsen, F.; Maj, C.; Fazaal, J.; Sivalingam, S.; Savchenko, O.; Mantri, A.; et al. TBK1 and TNFRSF13B Mutations and an Autoinflammatory Disease in a Child with Lethal COVID-19. NPJ Genomic Med. 2021, 6, 55, doi:10.1038/s41525-021-00220-w.
  10. Janezic, S.; Mahnic, A.; Kuhar, U.; Kovač, J.; Jenko Bizjan, B.; Koritnik, T.; Tesovnik, T.; Šket, R.; Krapež, U.; Slavec, B.; et al. SARS-CoV-2 Molecular Epidemiology in Slovenia, January to September 2021. Eurosurveillance 2023, 28, doi:10.2807/1560-7917.ES.2023.28.8.2200451.
  11. Šelb, J.; Bitežnik, B.; Bidovec Stojković, U.; Rituper, B.; Osolnik, K.; Kopač, P.; Svetina, P.; Cerk Porenta, K.; Šifrer, F.; Lorber, P.; et al. Immunophenotypes of Anti-SARS-CoV-2 Responses Associated with Fatal COVID-19. ERJ Open Res. 2022, 8, 00216–02022, doi:10.1183/23120541.00216-2022.
  12. The COVID-19 Host Genetics Initiative The COVID-19 Host Genetics Initiative, a Global Initiative to Elucidate the Role of Host Genetic Factors in Susceptibility and Severity of the SARS-CoV-2 Virus Pandemic. Eur. J. Hum. Genet. 2020, 28, 715–718, doi:10.1038/s41431-020-0636-6.
  13. Martin, A.R.; Williams, E.; Foulger, R.E.; Leigh, S.; Daugherty, L.C.; Niblock, O.; Leong, I.U.S.; Smith, K.R.; Gerasimenko, O.; Haraldsdottir, E.; et al. PanelApp Crowdsources Expert Knowledge to Establish Consensus Diagnostic Gene Panels. Nat. Genet. 2019, 51, 1560–1565, doi:10.1038/s41588-019-0528-2.
  14. The ACMG Laboratory Quality Assurance Committee; Richards, S.; Aziz, N.; Bale, S.; Bick, D.; Das, S.; Gastier-Foster, J.; Grody, W.W.; Hegde, M.; Lyon, E.; et al. Standards and Guidelines for the Interpretation of Sequence Variants: A Joint Consensus Recommendation of the American College of Medical Genetics and Genomics and the Association for Molecular Pathology. Genet. Med. 2015, 17, 405–423, doi:10.1038/gim.2015.30.
  15. Golinelli, D.; Boetto, E.; Maietti, E.; Fantini, M.P. The Association between ABO Blood Group and SARS-CoV-2 Infection: A Meta-Analysis. PloS One 2020, 15, e0239508, doi:10.1371/journal.pone.0239508.
  16. Liu, N.; Zhang, T.; Ma, L.; Zhang, H.; Wang, H.; Wei, W.; Pei, H.; Li, H. The Impact of ABO Blood Group on COVID-19 Infection Risk and Mortality: A Systematic Review and Meta-Analysis. Blood Rev. 2020, 100785, doi:10.1016/j.blre.2020.100785.
  17. Di Maria, E.; Latini, A.; Borgiani, P.; Novelli, G. Genetic Variants of the Human Host Influencing the Coronavirus-Associated Phenotypes (SARS, MERS and COVID-19): Rapid Systematic Review and Field Synopsis. Hum. Genomics 2020, 14, 30, doi:10.1186/s40246-020-00280-6.
  18. Elhabyan, A.; Elyaacoub, S.; Sanad, E.; Abukhadra, A.; Elhabyan, A.; Dinu, V. The Role of Host Genetics in Susceptibility to Severe Viral Infections in Humans and Insights into Host Genetics of Severe COVID-19: A Systematic Review. Virus Res 2020, doi:10.1016/j.virusres.2020.198163.
  19. Ramos-Lopez, O.; Daimiel, L.; Ramírez de Molina, A.; Martínez-Urbistondo, D.; Vargas, J.A.; Martínez, J.A. Exploring Host Genetic Polymorphisms Involved in SARS-CoV Infection Outcomes: Implications for Personalized Medicine in COVID-19. Int. J. Genomics 2020, 2020, 6901217, doi:10.1155/2020/6901217.
  20. Pairo-Castineira, E.; Rawlik, K.; Bretherick, A.D.; Qi, T.; Wu, Y.; Nassiri, I.; McConkey, G.A.; Zechner, M.; Klaric, L.; Griffiths, F.; et al. GWAS and Meta-Analysis Identifies 49 Genetic Variants Underlying Critical COVID-19. Nature 2023, 617, 764–768, doi:10.1038/s41586-023-06034-3.
  21. Severe Covid-19 GWAS Group; Ellinghaus, D.; Degenhardt, F.; Bujanda, L.; Buti, M.; Albillos, A.; Invernizzi, P.; Fernández, J.; Prati, D.; Baselli, G.; et al. Genomewide Association Study of Severe Covid-19 with Respiratory Failure. N. Engl. J. Med. 2020, 383, 1522–1534, doi:10.1056/NEJMoa2020283.
  22. Liu, P.; Fang, M.; Luo, Y.; Zheng, F.; Jin, Y.; Cheng, F.; Zhu, H.; Jin, X. Rare Variants in Inborn Errors of Immunity Genes Associated With Covid-19 Severity. Front. Cell. Infect. Microbiol. 2022, 12, 888582, doi:10.3389/fcimb.2022.888582.
  23. Khadzhieva, M.B.; Gracheva, A.S.; Belopolskaya, O.B.; Kolobkov, D.S.; Kashatnikova, D.A.; Redkin, I.V.; Kuzovlev, A.N.; Grechko, A.V.; Salnikova, L.E. COVID-19 Severity: Does the Genetic Landscape of Rare Variants Matter? Front. Genet. 2023, 14, 1152768, doi:10.3389/fgene.2023.1152768.

Round 2

Reviewer 3 Report

Comments and Suggestions for Authors

The authors answered all my questions satisfactorily. I think it's a good work to publish.